# Functional and Pharmacological Analyses of the Role of *Penicillium digitatum* Proteases on Virulence

**DOI:** 10.3390/microorganisms7070198

**Published:** 2019-07-12

**Authors:** Ana-Rosa Ballester, Mario López-Pérez, Beatriz de la Fuente, Luis González-Candelas

**Affiliations:** Department of Food Biotechnology, Institute of Agrochemistry and Food Technology, Spanish National Research Council (IATA-CSIC), Calle Catedrático Agustín Escardino 7, 46980 Paterna, Valencia, Spain

**Keywords:** *Agrobacterium tumefaciens* mediated transformation, citrus fruit, metal ion chelators, fruit–fungal interaction, transcription factor, virulence, protease inhibitors

## Abstract

*Penicillium digitatum* is the major postharvest pathogen of citrus fruit under Mediterranean climate conditions. Previous results have shown that proteases is the largest enzyme family induced by *P. digitatum* during fruit infection. In the present work, we addressed the study of the role of *P. digitatum*’s proteases in virulence following two complementary approaches. In the first approach, we undertook the functional characterization of the *P. digitatum*
*prtT* gene, which codes for a putative transcription factor previously shown to regulate extracellular proteases in other filamentous fungi. Deletion of *prtT* caused a significant loss in secreted protease activity during *in vitro* growth assays. However, there was no effect on virulence. Gene expression of the two major secreted acid proteases was barely affected in the Δ*prtT* deletant during infection of citrus fruit. Hence, no conclusion could be drawn on the role of these secreted acidic proteases on the virulence of *P. digitatum*. In the second approach, we studied the effect of different protease inhibitors and chelators on virulence. Co-inoculation of citrus fruit with *P. digitatum* conidia and a cocktail of protease inhibitors resulted in almost a complete absence of disease development. Analysis of individual inhibitors revealed that the metalloprotease inhibitor, 1,10-phenanthroline, was responsible for the observed effect. The application of metal ions reverted the protective effect caused by the metallopeptidase inhibitor. These results may set the basis for the development of new alternative treatments to combat this important postharvest pathogen.

## 1. Introduction

Proteases, also denoted as peptidases, proteinases, or proteolytic enzymes, can be classified according to the nature of the functional group at the active site. Most proteases belong to one of the four major families: Aspartic, cysteine, metallo, and serine peptidases. They are widely used in biotechnology, mainly in the food, leather, and detergent industries, in ecological bioremediation processes, and to produce therapeutic peptides [1]. They comprise a large number of proteins that account for a significant proportion of an organism’s gene count. Thus, species in the genera, *Aspergillus* or *Penicillium*, contain more than 200 and 100 annotated genes encoding for putative proteases in the MEROPS database (https://www.ebi.ac.uk/merops/), respectively. These enzymes play a major role in the physiology, morphogenesis, and metabolism of fungi. Their production is regulated in response to environmental signals, such as extracellular pH and carbon and nitrogen sources [2]. Proteases secreted into the environment play a crucial role in nutrition because they are needed for external digestion of macromolecular nutrients. In addition to nutrient utilization, microbial proteases are involved in many physiological processes, such as morphogenesis, germination, and conidial discharge [3]. Proteases play an important role in the mechanism of the virulence of pathogens by participating in the penetration and dissemination within the host, as well as by combating the host’s defense mechanisms [4,5,6,7]. The role of fungal proteases in plant infection has been less characterized than that of bacterial and animal pathogens. For example, *Sclerotinia sclerotiorum* produced aspartyl proteases, non-aspartyl acidic proteases, and serine proteases during infection of sunflower, and the increase of protease production was correlated with intensive colonization and maceration of the host tissues [8]. A UV-induced mutant of the tomato pathogen *Colletotrichum coccodes* defective in extracellular protease activity was unable to infect tomato fruits, although it showed normal vegetative growth and cellulase activity [9]. In *Fusarium oxysporum* f. sp. *lycopersici*, the synergistic action of a serine protease, FoSep1, and a metalloprotease, FoMep1, was required for cleavage and removal of the chitin-binding domain (CBD) from two tomato CBD-chitinases [10]. In addition, mutants of *F. oxysporum* f. sp. *lycopersici* lacking both FoSep1 and FoMep1 exhibited reduced virulence on tomato, confirming that secreted fungal proteases are important virulence factors by targeting CDB-chitinases to compromise an important component of the plant’s basal defense [10]. Fungalysins are a conserved family of metalloproteases in fungi and their role as chitinase-degrading enzymes has been demonstrated in *Colletotrichum graminicola*. The absence of the fungalisyn metalloprotease-encoding *CgfI* gene delayed fungal development during the infection process on maize leaves and, in parallel, maize leaves exhibit increased chitinase activity, suggesting that the fungus employs a CgfI-mediated strategy to control chitin signaling [11]. *Botrytis cinerea* is a typical necrotroph that secretes aspartic proteases during infection on various plant tissues. However, single or double deletant mutants in five genes encoding aspartic proteases did not result in any defect in virulence [12]. 

PrtT is a fungal-specific transcription activator of extracellular proteases that was first isolated and characterized in *Aspergillus niger* [13]. It is present in several *Aspergilli* and *Penicillia*, but absent in the genome of *Aspergillus nidulans* [14]. This transcription factor belongs to the fungal-specific Gal_4_-like Zn_2_Cys_6_ binuclear cluster protein family and plays an important role in the production of secreted proteases. Disruption of *prtT* in *A. niger* resulted in transformants unable to form a protease degradation halo on plates containing skim milk [13]. Moreover, an *Aspergillus oryzae prtT* disruption mutant produced lower levels of the alkaline serine protease S8 (AlpA) and to a lesser extent, the neutral metalloprotease M36 (NpI) compared to the wild type, confirming the role of PrtT in the regulation of the major proteases in this fungus [13]. Unexpectedly, microarray analysis revealed that the expression of genes involved in iron uptake and ergosterol synthesis was dramatically decreased in the *Aspergillus fumigatus* Δ*prtT* mutant, together with an upregulation of different secondary metabolite clusters [15]. However, in two independent works, this transcription factor was found to be not essential for virulence in this human opportunistic fungal pathogen, suggesting that either residual protease activity is sufficient to enable virulence or that proteases are dispensable for pathogenicity in this fungus [14,16]. Regarding the genus *Penicillium*, PrtT has been only characterized in *Penicillium oxalicum*. A transcription profiling analysis using RNA-Seq showed that many putative peptidase-encoding genes were either up- or down-regulated in a *P. oxalicum* Δ*prtT* mutant, including both secreted and intracellular proteases [17], confirming that PrtT is a global regulator of proteases. In addition, this transcriptomic study found that PrtT putatively regulates the transcription of specific amylases and major facilitator superfamily (MFS) transporters involved in the transport of nutrients, and of specific transporters and enzymes involved in lignocellulose degradation in response to nutrient limitation.

*Penicillium digitatum* is the most important postharvest pathogen of citrus fruit grown under Mediterranean conditions. It is a necrotrophic fungus that requires wounds in the fruit peel to penetrate and colonize the fruit tissue mostly through the deployment of maceration enzymes. The genome of this fungus contains 275 putative carbohydrate-active enzymes (CAZymes) assigned mostly to glycoside hydrolases, carbohydrate esterases, and polysaccharide lyases, among others, and to a lesser extent, to enzymes related to the degradation of cellulose and hemicellulose [18]. In comparison with other *Penicillium* spp., *P. digitatum* is enriched in polygalacturonases and pectinesterases, both involved in pectin degradation. This necrotrophic fungus possesses a small secretome compared to *Penicillium expansum* or *Penicillium italicum*, and proteases constitute a large proportion of its secretome [19]. The genome of *P. digitatum* encodes 119 proteases and 29 non-peptidase homologs (MEROPS peptidase database for *P. digitatum*, release 12.1, April 2019) [20]. The most abundant category corresponds to the superfamily of serine proteases, followed by metallo and cysteine proteases. In a previous study, we observed that genes coding for putative fungal proteases and plant cell wall-degrading enzymes represent the largest categories during the orange–*P. digitatum* interaction, with five secreted protease-encoding genes being among the most highly expressed genes during fruit infection [21]. In this report, we aim to analyze the role of *P. digitatum* proteases on virulence. In view of the large number of secreted proteases, we focused on PrtT, which regulates extracellular proteases, with the aim of reducing the production of secreted proteases as much as possible while avoiding the gene compensation effects observed when eliminating a single member of a large gene family. If the major proteases were regulated by PrtT, knocking out the corresponding gene could offer an alternative to study to the role of *P. digitatum* proteases in virulence. For this purpose, we followed a functional approach by constructing and characterizing a deletion mutant of the *prtT* gene. In addition, we undertook a pharmacological approach by using a set of protease inhibitors during the infection of citrus fruit by *P. digitatum*. Our results showed that 1,10-phenanthroline, a metalloprotease inhibitor, is able to control the development of *P. digitatum* in citrus fruit.

## 2. Materials and Methods

### 2.1. Fungal Strains and Growth Conditions

*Penicillium digitatum* (Pers.:Fr.) Sacc. strain Pd1 (PDIP, deposited at the Spanish Type Culture Collection with accession code CECT20795) was isolated from an infected grapefruit [18]. To prepare conidial suspensions, the strain was grown on potato-dextrose-agar (PDA) at 24 °C for 7 days. Conidia were scraped off the agar with a sterile spatula, suspended in sterile distilled water, and filtered through a nylon mesh. Conidia concentration was determined with a hemocytometer.

### 2.2. Generation and Verification of P. digitatum prtT Mutants

A BlastP search with the sequence of PrtT from *A. niger* (accession number XM_001402018.2) as the query was performed against the *P. digitatum* Pd1 proteome [18]. To construct the *prtT* gene replacement plasmid, 1.8 kb upstream and downstream flanking fragments of the *prtT* gene (PDIP_25240) were amplified from genomic DNA of *P. digitatum* (Pd1/PDIP), using the specific primers O1, O2, A3, and A4 (Table 1). These primers include vector-specific 9 bp long overhangs containing a single 2-deoxyuridine nucleoside in the 5’ end, which ensured directionality in the cloning reaction. The two flanking fragments were introduced into pRF-HU2 following the USER (uracil-specific excision reagent) protocol described by Frandsen et al. [22]. The resulted plasmid (denoted as pDprtT) was introduced into *Escherichia coli* DH5α chemical competent cells. Kanamycin-resistant transformants were screened by PCR for the presence of the promoter and the terminator with primer pairs RF1/RF6 and RF2/RF5, respectively (Table 1). Proper fusions were further confirmed by DNA sequencing and then the plasmid was transferred to *Agrobacterium tumefaciens* AGL1 electrocompetent cells. Transformation of *P. digitatum* Pd1 was done as previously described [18]. Equal volumes of induced bacterial culture and conidial suspension of *P. digitatum* strain Pd1 (10^5^ conidia/mL) were mixed and spread onto filter papers, which were placed on agar plates containing the co-cultivation medium. After co-cultivation at 24 °C for 48 h, the membranes were transferred to PDA plates containing 100 μg/mL of hygromycin B (InvivoGen, San Diego, CA, USA), as the selection agent for fungal transformants, and 200 μg/mL of cefotaxime (Calbiochem, San Diego, CA, USA) to inhibit the growth of *A. tumefaciens* cells. Hygromycin-resistant colonies appeared after 4 to 5 days of incubation at 24 °C. To ensure correct deletion of the *prtT* gene and the absence of ectopic insertions, conventional PCR and quantitative PCR (qPCR) were used to determine the gene copy number of the T-DNA inserted in *P. digitatum*. Firstly, disruption of the *prtT* gene was confirmed by PCR analyses of the transformants. Integration of the T-DNA by homologous recombination was examined using primer pairs HPHTER2/1F and HPHRO4/4R (Table 1) for the promoter and the terminator regions, respectively. Further verification of deletion of the target gene and the insertion of the hygromycin marker was done with primers 5F/6R followed by digestion with *EcoR*I. To determine the number of T-DNA molecules that had been integrated into the genome of each selected transformant, a qPCR analysis was carried out following an already demonstrated methodology described by several authors [21,23,24], using Pd1 DNA as the control. A primer pair (7F/8R) was designed within the T-DNA in the terminator region of the target gene, close to the selection marker. The *P. digitatum* actin gene (PDIP_18200) was chosen as a reference using the primer pair PdACTFor2/PdACTRev2 (Table 1). qPCR reactions were performed in a LightCycler480 System (Roche Diagnostics, Basel, Switzerland) using SYBR Green to monitor DNA amplification. For each primer pair and each sample, the PCR efficiencies (E) and the quantification cycle (Cq) were assessed using the LinRegPCR software version 2017.1 [25]. The number of T-DNA copies that were integrated in the genome of the transformants was calculated according to the formula: Copy number = (E_target gene_)^ΔCq_target gene_^(wild type – transformant)^/(E_reference gene_) ^ΔCq_reference gene_^(wild type – transformant)^ based on Pfaffl [26], which depends on E and the Cq value of the transformant versus the wild-type strain, and normalized in comparison to a reference gene that is present with the same copy number in both wild-type and transformant strains.

### 2.3. Characterization of the ΔprtT Knockout Mutants

For growth assessment and sporulation quantification, PDA plates were inoculated centrally with 5 μL of a conidia suspension (10^5^ conidia/mL) of the *P. digitatum* parental strain Pd1, the ectopic *prtT* mutant, and two Δ*prtT* knockout mutants. Cultures were incubated at 24 °C for up to 7 days. Mycelial growth was determined by measuring two perpendicular diameters of the growing colonies at day 7 after inoculation. Sporulation assessment was carried out by scraping the surface of the 7-day-old cultures with a spatula. Conidia concentration was measured by using a haemocytometer.

Proteolytic activity on solid medium was assessed based on Ward [27]. Spores (10^5^ conidia/mL) were inoculated onto filter discs overlaid on solid complete medium plates (PDA) containing a colony restrictor (2 mg/mL dichloran). After 4 days of incubation at 24 °C, the filters were removed and the plates were overlaid with a layer of skim milk agarose (1% agarose, 1% skim milk, 0.45% CaCl_2_, 0.6% acetic acid, pH 5.5), and milk clotting was allowed to proceed at 37 °C for 3 days. The extent of clotting was proportional to the number of proteases secreted by the colony that had occupied that position on the plate.

### 2.4. Chemicals

A protease inhibitor cocktail containing 1.4 mM of trans-Epoxysuccinyl-L-leucylamido(4-guanidino)butane (E-64), 500 mM of 1,10-phenanthroline, 100 mM of 4-(2-aminoethyl)benzenesulfonyl fluoride hydrochloride (AEBSF), and 2.2 mM of pepstatin A was purchased from Sigma-Aldrich (P8215) (St. Louis, MO, USA). AEBSF, bestatin hydrochloride, E-64, phosphoramidon disodium salt, pepstatin A, ferrozine, diethyldithiocarbamate, and 1,10-phenanthroline hydrochloride monohydrate were also purchased from Sigma-Aldrich. Ethylenediaminetetraacetic acid calcium disodium salt dehydrate (EDTA) and dimethylsulfoxide (DMSO) were obtained from Applichem (Darmstadt, Germany).

### 2.5. Orange Fruit Infection Assays

To analyze the role of *prtT* in the pathogenicity of *P. digitatum*, we artificially inoculated the parental strain Pd1, one ectopic mutant (e*prtT*3), and two knockout mutants (Δ*prtT*44 and Δ*prtT*70) on sweet oranges: ‘Navelate’ and ‘Lane late’ mature oranges that were obtained from a packinghouse in Lliria, Valencia (Spain) the same day of harvesting before receiving any postharvest treatment. They were brought to the laboratory, surface-disinfected with 5% sodium hypochlorite for 5 min, rinsed with tap water, and allowed to dry. The next day, oranges were wounded four times around the equator with a nail (3 mm in depth) and were immediately inoculated by adding 10 μL of a conidial suspension (10^4^ conidia/mL). Three replicates of five infected fruits with four wounds per fruit were placed on plastic boxes and incubated at 20 °C and 90% relative humidity for 7 days. Disease incidence (measured as the percentage of infection) and severity (as maceration diameter, in mm) were determined at day 5 and 7 post inoculation (dpi). Analysis of variance was performed to test the different incidence among strains at 5 dpi. Means were separated using the Tukey test with *p* < 0.05, using Statgraphics Stratus (Statgraphics Technologies, Inc., The Plains, VA, USA).

To study the effect on virulence of either the protease inhibitor cocktail, its individual components, and other different protease inhibitors and chelators, *P. digitatum* conidia were artificially co-inoculated with the proteinase inhibitor cocktail, E-64, 1,10-phenanthroline, AEBSF, pepstatin A, the double or triple combination of the different components, and with bestatin, phosphoramidon, EDTA, EGTA, ferrozine, and DETC in mature oranges as described above. The assayed concentration of each compound is indicated in the figure legend. Incidence and severity were measured up to 7 dpi.

The effect of metals and the chelator 1,10-phenanthroline was assayed by co-inoculation of 10^4^ conidia/mL of *P. digitatum* Pd1 with different metal ions (ZnSO_4_, CuSO_4_, MnSO_4_, and FeSO_4_) at 0.5 mM either in the presence or absence of 1,10-phenanthroline 0.5 mM in mature oranges as described previously. Disease incidence was determined at 4, 5, and 6 dpi.

### 2.6. Gene Expression Analysis

For RNA extraction, mature oranges were wounded using a nail and inoculated with 10 μL of a conidial suspension (10^6^ conidia/mL, 16 wounds per fruit) from either the *P. digitatum* parental strain Pd1, the ectopic mutant (e*prtT*3), or a knockout mutant (Δ*prtT*70). Inoculated fruits were stored at 20 °C and high humidity for 24, 48, and 72 h. After each storage time, cylinders of peel containing the flavedo and the albedo of the fruit were removed using a cork borer of 5 mm centered in the inoculation point. Each biological replicate consisted of 80 discs (16 wounds per 5 fruits) and three biological replicates were collected at each sampling time point. All samples were immediately frozen in liquid nitrogen and then ground to a fine powder for subsequent RNA extraction. Spores of the parental strain, the ectopic mutant, and the knockout mutant were also frozen for subsequent RNA extraction.

Total RNA extraction from *P. digitatum* spores and from macerated orange peel tissue was done following a previously published protocol [28] with minor modifications. One gram of frozen tissue was extracted with 10 mL of RNA extraction buffer (100 mM Tris HCl pH 8.0, 100 mM LiCl, 10 mM EDTA pH 8.0, 1% SDS, 1% PVP-40, and 1% β-mercaptoethanol). After phenol extraction, total nucleic acids were precipitated by adding one-tenth volume of 3M sodium acetate, pH 5.2, and two volumes of cold ethanol, and incubating at −20 °C for at least 30 min. For non-macerated orange peel tissue, RNA extraction was done according to López-Pérez et al. [21]. RNA concentration was measured spectrophotometrically. DNase treatment and first-strand cDNA synthesis were conducted with the Maxima H Minus cDNA synthesis kit with dsDNase (Thermo Scientific, Waltham, MA, USA) using 2 μg of total RNA according to the manufacturer’s instructions. RT-qPCR was conducted following the MIQE (Minimum Information for Publication of Quantitative Real-Time PCR Experiments) guidelines [29]. Gene-specific primer sets (Table 1) were designed for gene expression analysis with Primer3Plus [30]. Real-time qPCR reactions were performed in a LightCycler480 System (Roche Diagnostics, Basel, Switzerland) using SYBR Green to monitor cDNA amplification. Gene expression measurements were derived from three biological replicates and two technical replicates. Relative gene expression (RGE) was calculated using the formula described by Pfaffl [26]. For each primer pair and each sample, the PCR efficiency (E) and the quantification cycle (Cq) were assessed using LinRegPCR software version 2017.1. Amplicon specificity was examined by analysis of the melting curve. The Cq value for the reference normalization factor (REF) was calculated by taking actin (PDIP_18200) as the reference gene, using primer pairs PdACTFor2/PdACTRev2 [21].

## 3. Results

### 3.1. Identification of a prtT Ortholog in P. digitatum and Construction of Knockout Mutants

In order to identify a PrtT homolog in *P. digitatum*, we interrogated the automatically annotated *P. digitatum* genome sequence available at the NCBI (National Center for Biotechnology Information) [18]. A BlastP search was performed using the sequence of PrtT from *A. niger* (XP_001402055.1) as a query. PDIP_25240 was the protein with the highest identity, 52%, regularly distributed along the whole sequence, although it missed about 300 aa at the N-terminus that were present in *A. niger* PrtT. The original automatic annotation of PDIP_25240 corresponded to a protein with 368 aa, much shorter than *A. niger* PrtT, which contains 623 aa. A more detailed examination of the *P. digitatum* Pd1 locus allowed the reannotation of the *prtT* gene, which contains five exons, as its *A. niger prtT* counterpart, and codes for a protein with 636 aa. Multiple sequence alignment of *P. digitatum* PrtT with orthologues from other *Aspergillus* and *Penicillium* species showed that they all contained the fungal-specific Zn_2_Cys_6_ binuclear DNA binding cluster domain, conserved in both genera (Figure 1A) [14]. The identities of *P. digitatum* PrtT (636 aa) with the amino acid sequences of PrtT from *Penicillium rubens* (GenBank accession number XP_002565177), *P. oxalicum* (EPS29021), *A. niger* (XP_001402055), and *A. fumigatus* (KEY83531) were 89%, 63%, 52%, and 50%, in 637, 629, 616, and 613 amino acids, respectively.

Targeted gene disruption of *prtT* using the methodology previously described [18] was performed to investigate the role of the encoded protein in the pathogenicity of *P. digitatum*. The first step of the gene deletion strategy was to construct the plasmid pDprtT using the USER-friendly cloning technique. Positive *E. coli* transformants were selected as kanamycin-resistant colonies and screened by PCR (data not shown). Afterward, the plasmid was introduced into electrocompetent *A. tumefaciens* AGL1 cells. The following step was the transformation of *P. digitatum* by co-cultivation with *A. tumefaciens*. Putative transformants were selected in the presence of hygromycin B.

The correct deletion of the *prtT* gene was verified by PCR using the primers 1F/HPHTer2 for the identification of the integration at the promoter region, and HPHPRO4/4R for the terminator region (Table 1 and Figure 1B). Figure 1C shows the expected band pattern for both knockout Δ*prtT*44 and Δ*prtT*70 mutants, not observed in the parental strain Pd1 or the ectopic e*prtT*3 mutant. The confirmation of the presence/absence of the *prtT* gene was also observed using primers 5F/6R, which flanks the coding region of the *prtT* gene, after digestion of the PCR amplification product with *EcoR*I. A DNA fragment of 2700 bp corresponding to the *prtT* gene was observed in the parental strain, whilst two fragments of 2150 and 750 bp corresponding to the replacement of the *prtT* gene by the hygromycin-resistance gene were observed in both knockout mutants. Thus, the hygromycin-resistance marker was integrated by double homologous recombination replacing the *prtT* gene, whereas the ectopic transformant e*prtT*3 showed the three bands. The number of T-DNA copies integrated into the genome of each transformant was assessed by qPCR analysis, confirming that the knockout mutants contained a single T-DNA integration (Table 2).

### 3.2. Characterization of P. digitatum ΔprtT Knockout Mutants

Conidia production of the parental strain Pd1, ectopic, and Δ*prtT* knockout mutants were assessed on PDA plates after 7 days of incubation. Fungal growth was assayed by measuring two perpendicular diameters of at least five independent colonies up to 7 days post inoculation, when conidia were collected to determine conidia production (Figure 2A). No differences were observed in colony growth among different strains (data not shown). The parental strain and the two knockout mutants produced a similar amount of conidia per colony area (approximately 215,000 conidia/mm^2^), whereas the ectopic mutant had the highest production (approximately 290,000 conidia/mm^2^).

Based on the hypothesis that the *prtT* deletion mutant should be impaired in the production of extracellular proteases, a protein degradation assay was performed with the four strains in solid culture medium. We inoculated the strains onto filter discs on PDA plates containing 2 mg/mL of dichloran to restrict colony growth. After growth of the colonies during 4 days at 24 °C, the filters were removed and the plates were overlaid with a layer of agarose containing skim milk. Hydrolysis of milk proteins, mostly casein, at 37 °C during 3 days led to the appearance of a proteolytic halo. A clearing zone was observed only with strains expressing the *prtT* gene: The parental strain Pd1 and the ectopic mutant (Figure 2B). No halo, or a very thin halo, was observed in both knockout mutants, indicating that no or poor proteolysis took place.

### 3.3. The P. digitatum ΔprtT Mutants Are Not Altered in Virulence 

To test the role of PrtT in the pathogenicity of *P. digitatum*, the virulence of the parental strain Pd1, the ectopic e*prtT*3 mutants and the two knockout Δ*prtT*44 and Δ*prtT*70 mutants was tested. We inoculated ‘Navelate’ (Figure 3A,B) and ‘Lane late’ (Figure 3C–F) sweet oranges in three independent experiments and measured the incidence (percentage of infection) and the disease severity (measured as the diameter of the macerated tissue) at 5 and 6 to 7 days post inoculation in three independent infection assays. The incidence and severity results showed that the knockout mutants were as virulent as the parental strain Pd1 in all three experiments. Only in one out of the three independent replicates (experiment 2, Figure 3C,D) was the incidence and maceration diameter lower in the ectopic mutant compared to the parental strain and the two knockout mutants. The multifactor analysis of variance at 5 dpi to test the significant interactions amongst the factors showed that there was not a difference among the fungal incidence of the four samples (*p*-value = 0.2291), but there was a statistical significance depending on the date of the experiment at the 95% confidence level (*p*-value = 0.0000).

### 3.4. PrtT Has Only a Minor Effect on the Expression of the Two Major Proteases during Infection

Previous results showed that genes coding for fungal proteases, plant cell-wall related enzymes, redox homeostasis, and detoxification processes were the major categories induced during the infection of citrus fruit by *P. digitatum* [21]. The most represented gene in the subtracted cDNA library was PDIP_82060 (denoted as *pep1*), which codes for a putative aspartic endopeptidase Pep1. In the same study, other genes coding for putative proteases/peptidases were also detected: An aspergillopepsin (PDIP_06020), a tripeptidyl peptidase (*aor1*; PDIP_12220), a carboxypeptidase (PDIP_71590), and a serine peptidase (PDIP_67670). According to SignalIP prediction (http://www.cbs.dtu.dk/services/SignalP/), four of them, except the serine peptidase, are extracellular proteases. As it was evident from the proteolytic assay that PrtT regulates the production of different extracellular proteases, we monitored the transcript levels of the *prtT* gene and the genes that encode the two major secreted proteases of *P. digitatum* during the infection process (PDIP_82060 and PDIP_12220). For this study, we collected spores and peel tissue of oranges infected with the parental strain Pd1, the ectopic e*prtT*3 mutant, and the knockout mutant Δ*prtT*70, at 12, 24, 48, and 72 hours post inoculation (hpi). A similar pattern of expression for *prtT* (PDIP_25240) was observed in both the parental strain Pd1 and the ectopic mutant (Figure 4A), showing the highest levels of expression in the spores. Expression of *prtT* decreased abruptly just during spore germination and remained very low during the infection process. As it was expected, no amplification of this gene was observed in the Δ*prtT*70 knockout mutant at any time point (Figure 4A). Results of the time-course experiment showed that both the putative aspartic endopeptidase *pep1* (PDIP_82060, Figure 4B) and the tripeptidyl peptidase *aor1* (PDIP_12220, Figure 4C) encoding genes showed a similar pattern of expression during the development of Pd1 in oranges, with the higher levels of expression of the gene encoding the aspartic endopeptidase. For both genes, maximum expression was observed at 48 hpi and, thereafter, its expression decreased. The expression of the two protease-encoding genes in the ectopic e*prtT*3 and the Δ*prtT*70 knockout mutant followed a pattern similar to that found in the parental strain, with maximum expression at 48 to 72 hpi. The differences were in the expression levels, which were lower in the knockout mutant.

### 3.5. Application of Protease Inhibitors Reduced the Virulence of P. digitatum in Citrus Fruit

In order to study the effect of different protease inhibitors on virulence, we co-inoculated citrus fruits with *P. digitatum* strain Pd1 conidia and a cocktail containing different protease inhibitors. Disease incidence and severity were determined up to 6 dpi. Results showed that the protease inhibitor cocktail at 1% was very effective (Figure 5), resulting in no visible disease development for up to 6 days, when the control fruits reached 98.3% infection by day 5. Even when applied at 0.1%, disease development was reduced by 66% at 5 dpi. This protection was not due to the presence of DMSO, the solvent used to prepare the cocktail, as disease development was not affected by the solvent at the final concentrations present in the co-inoculation mixtures. The components of the cocktail were 100 mM AEBSF (a serine protease inhibitor), 500 mM 1,10-phenanthroline (a metalloprotease inhibitor), 2.2 mM pepstatin A (a specific inhibitor of aspartyl proteases), and 1.4 mM E-64 (a broad-spectrum cysteine-protease inhibitor). We then conducted infection assays with the individual inhibitors at two different conidia concentrations, 10^4^ and 10^5^ conidia/mL (Figure 5A,B). Disease incidence of *P. digitatum* in artificially inoculated oranges was 96.7% and 100% after 5 dpi with 10^4^ and 10^5^ conidia/mL, respectively (Figure 5A,B). Similar results were observed with the co-inoculation of *P. digitatum* with the individual inhibitors, E-64, AEBSF, and pepstatin A. However, a complete absence of disease development up to 6 days was observed when *P. digitatum* was co-inoculated with either 1% cocktail or 10 mM 1,10-phenanthroline. These results clearly showed that 1,10-phenanthroline was the inhibitor with a major role inhibiting disease development. To further evaluate the possible interaction of 1,10-phenanthroline with the other inhibitors present in the cocktail, we co-inoculated *P. digitatum* with all possible combinations of the four protease inhibitors (Figure 5C,D). Our results showed that only the mixtures containing 1,10-phenanthroline were able to control the infection of Pd1 in oranges, confirming that this compound was responsible for the reduction of the *P. digitatum* infection in citrus fruit.

### 3.6. Application of Metalloproteinase Inhibitors and Chelators

Previous results have shown that 1,10-phenanthroline was the compound involved in the reduction of *P. digitatum* infection in mature orange fruits. It is known that this compound is an inhibitor of metalloproteases by removal and chelation of the metal ions required for enzyme activity. In order to study the effect of other metalloprotease inhibitors and chelators in the pathogenicity of *P. digitatum*, we co-inoculated the fungus with different known inhibitors and chelators: 1,10-phenanthroline, bestatin (an aminopeptidase inhibitor), phosphoramidon (a metalloendopeptidase inhibitor), and the metal ion chelators EDTA, EGTA, ferrozine, and DETC (Figure 6A). Neither protease inhibitors, bestatin or phosphoramidon, nor EGTA, DETC, or ferrozine reduced significantly the percentage of infection of *P. digitatum* compared to the parental strain Pd1, with EDTA being the only chelator that significantly reduced disease development. Only the co-inoculation of Pd1 with 1,10-phenanthroline completely prevented the development of the fungus in the oranges.

To test the hypothesis that the activity of 1,10-phenanthroline is related to the chelation of metal ions, we co-inoculated *P. digitatum* Pd1 with four different metal ions (ZnSO_4_, CuSO_4_, MnSO_4_, and FeSO_4_) either alone or in the presence of 1,10-phenanthroline (Figure 6B). The co-inoculation of *P. digitatum* with the four different metal ions did not deter the development of the pathogen; however, as we indicated previously, the application of 1,10-phenanthroline substantially reduced disease incidence. The combined application of 1,10-phenanthroline with CuSO_4_ partially reverted the effect of the protease inhibitor, and only the application of ZnSO_4_ and FeSO_4_ together with 1,10-phenanthroline totally reverted the effect of the tested protease inhibitor, confirming the hypothesis that 1,10-phenanthroline acts by chelating metal ions that are necessary for the development of *P. digitatum* in oranges.

## 4. Discussion

This study aimed to characterize the role of secreted proteases in the virulence of *P. digitatum* towards citrus fruit. Because most of the protease-encoding genes belong to gene families containing an elevated number of members, it is not technically feasible to delete more than a few of these genes at a time. This methodology has been described in *B. cinerea* by constructing single and double knockout mutants of five members from an aspartic proteinase gene family; however, the role of them in the virulence is not completely clear [12]. In our study, instead of simultaneously deleting several *P. digitatum* protease genes at a time, we designed two alternative approaches to determine the contribution of secreted proteolytic activities to the virulence of *P. digitatum*: (i) Construction of *prtT* knockout mutants and characterization of the mutants during *in vitro* and *in vivo* growth, and (ii) the application of different protease inhibitors during the infection of *P. digitatum* in sweet oranges.

As a first approach, we focused on the *P. digitatum prtT* gene, which encodes a putative transcription factor controlling the expression of multiple secreted proteases. The disruption of the *prtT* gene has been previously described in *A. niger* [13], *A. fumigatus* [14,15,16,31], and *P. oxalicum* [17]. As far as we are aware, there are no reports on the possible role of secreted proteases in the virulence of fungal pathogens of citrus fruit. Characterization of two independent knockout mutants revealed that PrtT is required for the production of several extracellular proteases by *P. digitatum* (Figure 2B). However, the absence of the regulator has just a small influence on the expression of the genes encoding the major putative extracellular proteases secreted by *P. digitatum* during the infection of sweet oranges (Figure 4) and has no effect in the virulence of the fungus (Figure 3).

Secreted protease activity depends on the pH of the growth media and the nitrogen or carbon source, among others [2]. For example, in *A. fumigatus*, protease activity was repressed by ammonia, or elevated pH, and activated in the presence of proteins as the sole nitrogen source [16]. In the present study, conidia production per area of growth of the *P. digitatum* Δ*prtT* knockout mutants were similar to those of the parental strain under the tested conditions (Figure 2A). It has been described that proteases constitute the largest group of *P. digitatum* genes up-regulated during the infection of oranges and that they might contribute to pathogenicity in different ways, such as degrading plant cell components or inactivating defense proteins [21]. We have shown that the protease activity of the Δ*prtT* knockout mutants grown on PDA and further incubated with skim milk was reduced to almost undetectable levels (Figure 2B) with respect to the parental strain and the ectopic mutant. This result indicates that PrtT is involved in the regulation of at least some proteases that are required by *P. digitatum* to degrade skim milk when the fungus grows on PDA medium. However, whether PrtT is involved in the regulation of additional proteases would require further experiments.

In order to determine the role of PrtT in virulence, the knockout Δ*prtT*70 mutant, the ectopic mutant, and the parental strain *P. digitatum* were artificially inoculated in sweet oranges (Figure 3). After 7 days of inoculation, no significant differences were observed in the incidence and the maceration diameter among them, suggesting that PrtT is not involved in the virulence of this postharvest pathogen in citrus fruit. Similar results have been previously observed in *A. fumigatus*, in which the *prtT* gene appears not to be essential for pathogenicity in animal models [14,16,31]. The *A. fumigatus* Δ*prtT* mutant showed reduced killing of lung alveolar cells and erythrocyte lysis [14,16]; however, the mutant strain showed wild-type virulence in infected neutropenic mice, suggesting that perhaps residual protease activity was sufficient to enable virulence [14,16]. Our results suggest that although PrtT regulates a group of secreted proteases (Figure 2B), it has no role in virulence (Figure 3). As already pointed out in the work done with *A. fumigatus* PrtT, this result could suggest that either residual protease activity is sufficient to enable virulence or that proteases are dispensable for pathogenicity in this fungus [14,16]. Another possible explanation is that the major extracellular proteases secreted by the pathogen during the infection process are not regulated by the *prtT* gene. To test this hypothesis, we analyzed the expression of the two genes coding for the major putative proteases during the *P. digitatum* infection process: The aspartic endopeptidase *pep1* encoding gene (PDIP_25240) and the tripeptidyl peptidase *aor1* encoding gene (PDIP_12220) [21]. The expression of these two genes during the infection of oranges was barely affected by the loss of the *prtT* gene, indicating that the regulation of these genes depends mostly on another factor(s). XprG is another transcription factor that regulates extracellular protease production in *Aspergillus nidulans*, a fungus that lacks a PrtT homolog. Deletion of both *A. fumigatus xprG* and *prtT* genes resulted in the generation of a mutant with almost no ability to degrade proteins; however, it retained wild-type virulence in murine systemic and pulmonary models of infection [31]. In the case of *P. digitatum*, we identified a single XprG ortholog by amino-acid similarity to *A. fumigatus* XrpG (data not showed). The possibility that these two major proteinases are relevant for *P. digitatum* virulence in sweet oranges and the role of XrpG in the regulation of protease secretion should be further explored. In future experiments, we might generate *P. digitatum* deletion mutants in these two genes encoding major extracellular proteases and in the *xrpG* putative transcription factor gene.

In the second approach, we investigated the effect of the application of different protease inhibitors on the virulence of *P. digitatum*. The presence of protease inhibitors has been described in plants and they are part of the pathogenesis-related proteins [32]. The first protease inhibitor proteins, trypsin and chymotrypsin inhibitors, with antifungal activity were described in *Brassica oleracea* by Lorito et al. [33], and subsequently, other protease-inhibitor proteins, such as cystatin, have been described in plants [34,35,36,37]. In the present study, we investigated the role of different protease inhibitors on the virulence of *P. digitatum* in oranges, and after 6 days post-inoculation, only 1,10-phenanthroline and the combinations containing this metalloprotease inhibitor were effective in controlling the development of *P. digitatum* in oranges. We tested other metalloprotease inhibitors, such as bestatin and phosphoramidon, and different metal ion chelators, such as EDTA, EGTA, ferrozine, and DETC (Figure 6A); although none of them were as effective as 1,10-phenanthroline in reducing the development of *P. digitatum* in citrus fruit, we observed some protective effect with some chelators, specially EDTA.

1,10-phenanthroline is a membrane permeable heterocyclic compound with the ability to sequester metal ions in biological systems, forming coordination compounds with them [38]. It has the capability of inhibiting the biological role of metal-dependent proteins, interfering with metal acquisition, bioavailability, and metabolism for crucial reactions; disturbing the microbial cell homeostasis; and culminating in the blockage of microbial nutrition, growth, development, and playing an important role in the *in vivo* infection progression [39]. The utilization of metal complexes containing 1,10-phenanthroline as antimicrobials against a broad spectrum of bacteria and as a potential alternative to antibiotics has been described previously [39]. Phenanthroline-based complexes can penetrate the cell membrane and can interact with relevant biomolecules in the microorganisms, leading to inhibition of the cell growth and causing cell death, exhibiting a broad spectrum of both antibacterial (e.g., against *E. coli* and *Pseudomonas aeruginosa*) and antifungal (e.g., against *A. niger* and *Fusarium solani*) activities. Metal sequestration is also found in nature as a means to combat microbial infection. The process by which a host organism sequesters trace minerals in an effort to limit pathogenicity during infection has been designated ‘nutritional immunity’ [40,41,42]. Well–studied examples of nutritional immunity include the production of the iron binding lactoferrin or the zinc and manganese binding protein calprotectin [40,41,42]. The antimicrobial activity of these proteins is mostly due to their capability to bind metal ions, as is the case of siderophores secreted by many biocontrol microbial antagonists [43,44]. Moreover, it has been hypothesized that the high level of gluconic acid secretion found during pathogenicity of apple fruits by *P. expansum* could be involved in the formation of iron chelates, which could favor iron acquisition and pathogenicity [45].

In the present work, we co-inoculated *P. digitatum* with some metal ions (ZnSO_4_, CuSO_4_, MnSO_4_, and FeSO_4_) in sweet oranges to further analyze the role of 1,10-phernatroline as a metalloproteinase inhibitor and as a chelator. We chose these four metal ions because 1,10-phenanthroline has a very high affinity for Fe^2+^, Zn^2+^, and Cu^2+^, but very low affinity for Mn^2+^. We hypothesized that the effect of these metal ions reverting the inhibition of 1,10-phenanthhroline would be related to their affinity to this chelator. The co-inoculation had no effect in the development of the pathogen, with an incidence of 100% of infected wounds after 7 dpi (Figure 6B). However, as indicated previously, the application of 1,10-phenanthroline drastically reduced the growth of the fungal pathogen in the fruit. The application of a plant protease inhibitor as an antifungal agent has been evidenced in transgenic rice constitutively expressing a potato carboxypeptidase inhibitor; these plants exhibit resistance against the economically important pathogens, *Magnaporthe oryzae* and *Fusarium verticillioides* [46]. The effect of 1,10-phenanthroline preventing the infection of citrus fruit by *P. digitatum* was partially reverted by application of CuSO_4_, and was completely reverted by the addition of ZnSO_4_ and FeSO_4_ (Figure 6B), indicating that the fungus is most susceptible to zinc and iron and, to a lesser extent, copper deprivation during the infection process. The concept of fungal micronutrient scavenging can be used in future studies aimed at developing a product containing a chelator, such as 1,10-phenanthroline, capable of reducing the development of fungal pathogens during postharvest.

## 5. Conclusions

By way of conclusion, this study showed that PrtT, a putative transcription factor that regulates extracellular proteases, is not the major factor affecting the regulation of the two major extracellular proteases secreted by *P. digitatum* during the infection of citrus fruits and that this gene is not involved in *P. digitatum* virulence. Furthermore, the good results in decay control obtained in the present study with 1,10-phenanthroline, a well-known metal chelator, warrants the exploration of a new possible target in fungal control: Metal chelation as a means to restrict micronutrient availability to pathogens.

## Figures and Tables

**Figure 1 microorganisms-07-00198-f001:**
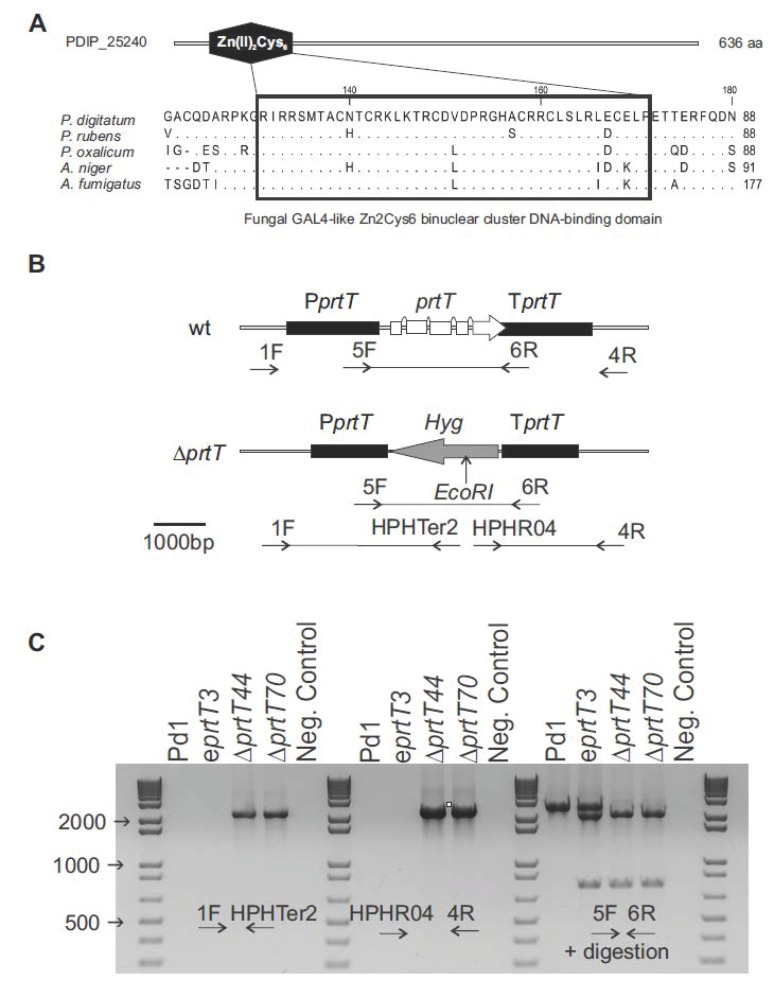
Deletion of *Penicillium digitatum prtT*. (**A**) A fungal-specific Zn_2_Cys_6_ binuclear DNA binding cluster (black box) located at the N-terminal region of *P. digitatum* PrtT. The GeneBank accession number of aligned homologs are as follows: *Penicillium rubens* Wisconsin 54-1255 (XP_002565177), *Penicillium oxalicum* 114-2 (EPS29021), *Aspergillus niger* CBS 513.88 (XP_001402055.1), and *Aspergillus fumigatus* var. RP-2014 (KEY83531). Identical amino acids are indicated by a dot ‘·’. (**B**) Diagram of the wild type *P. digitatum* Pd1 and deleted *prtT* loci, including the diagnostic primers used for checking the deletion (see also Table 1). P*prtT*: *P. digitatum prtT* gene promoter; T*prtT*: *P. digitatum prtT* gene terminator; Hyg; hygromycin resistance gene. (**C**) Polymerase chain reaction amplification of the wild type Pd1, the ectopic e*prtT*3 mutants, and the two knockout Δ*prtT*44 and Δ*prtT*70 mutants with diagnostic primers.

**Figure 2 microorganisms-07-00198-f002:**
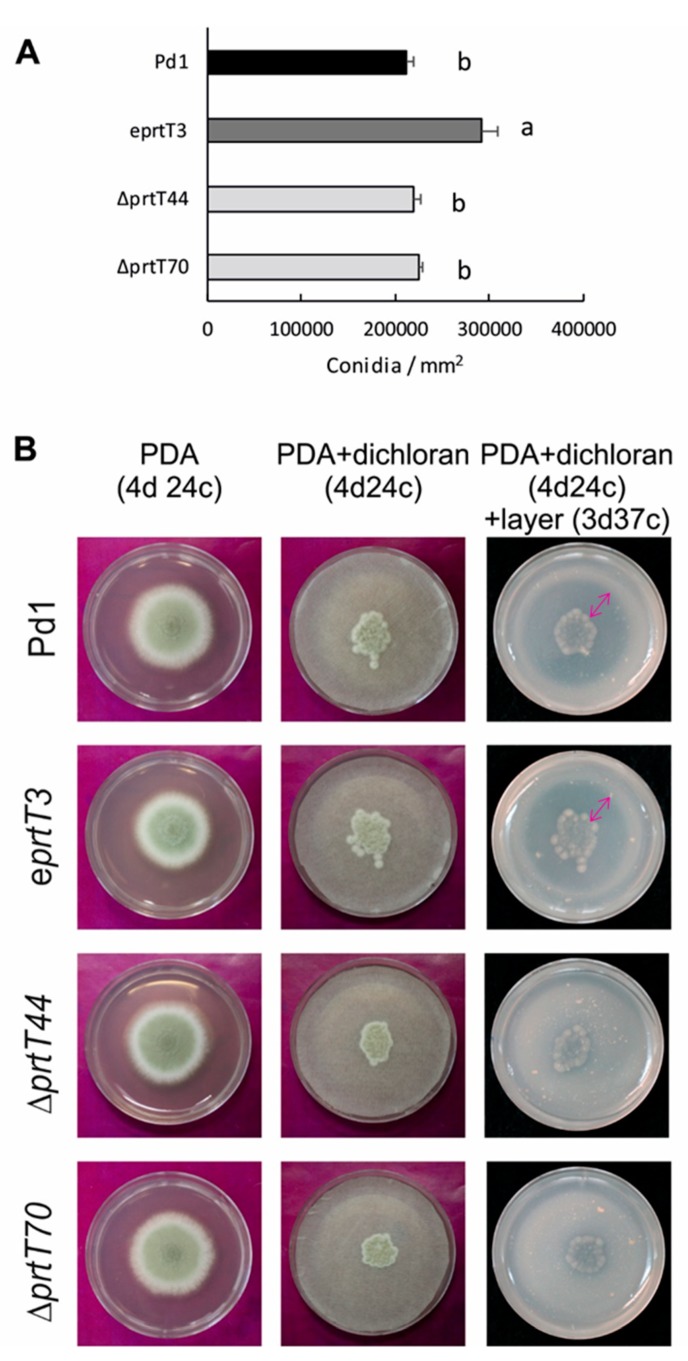
Phenotypic analysis of the Δ*prtT* mutants. (**A**) Conidia production per colony area (in mm^2^) of the parental strain Pd1, the ectopic e*prtT*3 mutant, and the two knockout mutants Δ*prtT*44 and Δ*prtT*70. Strains were grown on PDA plates and the conidia were collected and countered after 7 days at 24 °C. Error bars indicate the standard deviation of at least three biological replicates. Different letters indicate statistically significant differences among strains at *p* < 0.05. (**B**) Colony view of the parental strain Pd1, the ectopic e*prtT*3 mutant, and the two knockout Δ*prtT*44 and Δ*prtT*70 mutants after growing on PDA during 4 days at 24 °C, or on PDA containing dichloran during 4 days at 24 °C, or on PDA + dichloran during 4 days at 24 °C followed by a 3 day incubation at 37 °C after adding an agarose layer containing skim milk. The Δ*prtT* mutants formed a reduced proteolytic halo (indicated with an arrow) on skim milk media.

**Figure 3 microorganisms-07-00198-f003:**
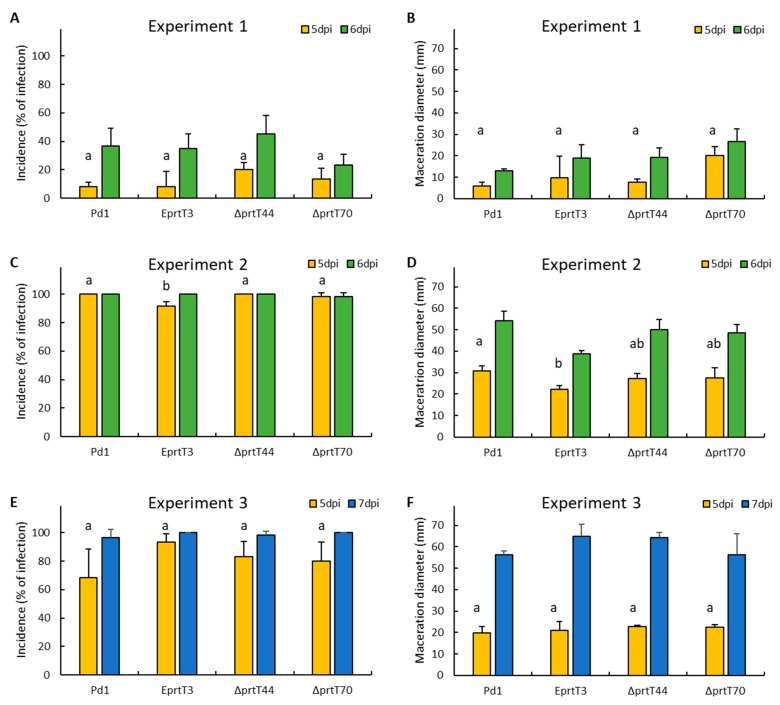
Three independent virulence assays of *P. digitatum* strain Pd1, the ectopic e*prtT*3 mutant, and the two knockout Δ*prtT*44 and Δ*prtT*70 mutants towards citrus fruit. Navelate (**A**,**B**) and Lane late (**C**–**F**) oranges were wounded and inoculated with 10 μL of a spore suspension (10^4^ conidia/mL) of the different strains. Fruits were incubated at 20 °C and 90% relative humidity for up to 7 days. There were three replicates of five fruits and four wounds per fruit. Bars show the mean values of the incidence as the percentage of infected wounds (**A**,**C**,**E**) and the maceration diameter in mm (**B**,**D**,**F**), and their standard deviations at 5 and 6 to 7 days post-inoculation (dpi). Bars labeled with the same letter at 5 dpi do not differ at the 95% confidence level based on the Tukey’s test.

**Figure 4 microorganisms-07-00198-f004:**
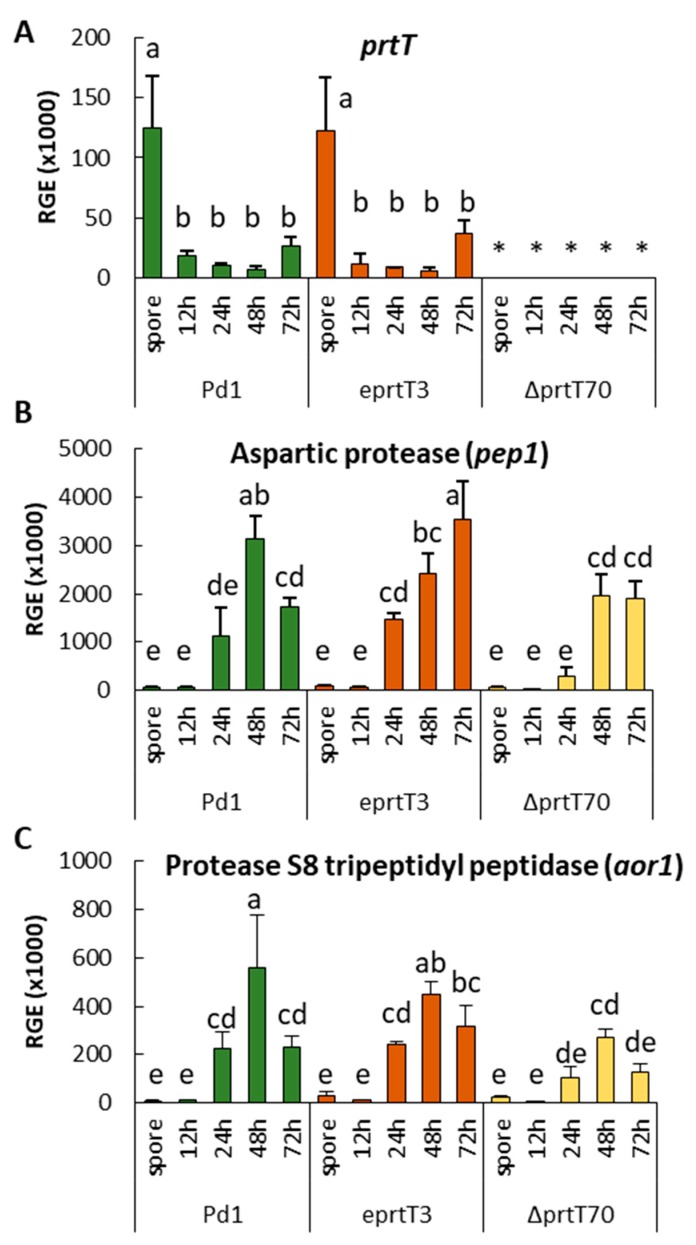
Analysis of normalized *in vivo* relative gene expression (RGE) in *P. digitatum* strain Pd1 (green bars), the ectopic e*prtT*3 mutant (orange bars), and the knockout Δ*prtT*70 mutant (yellow bars). The analysis was carried out in spores and at 12, 24, 48, and 72 hours after inoculation of orange fruits with the different strains. Bars show the mean values of three biological replicates and their standard deviations. The gene expression is relative to the *P. digitatum* actin gene as a reference. (**A**) PDIP_25240: Transcriptional activator of proteases, *prtT*; (**B**) PDIP_82060: aspartic protease, *pep1*; and (**C**) PDIP_12220: putative protease S8 tripeptidyl peptidase I, *aor1*. Bars labeled with the same letter indicate that there are no statistically significant differences at the 95% confidence level based on the Tukey’s test. The asterisk * indicates no expression level detected under the tested conditions.

**Figure 5 microorganisms-07-00198-f005:**
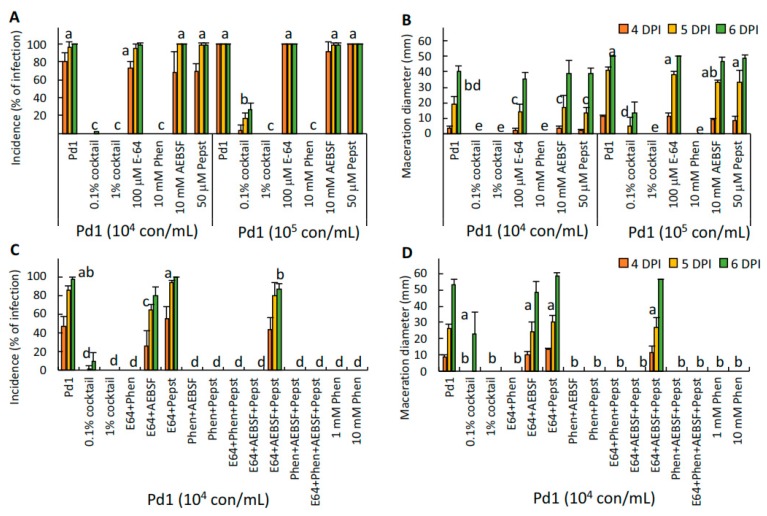
Disease incidence (percentage of infection, **A** and **C**) and severity (maceration diameter, in mm, **B** and **D**) caused by *P. digitatum* strain Pd1 in artificially inoculated oranges in two independent experiments (experiment 1: **A** and **B**; experiment 2: **C** and **D**) after 4, 5, and 6 days post infection (dpi). Conidia were co-inoculated with the proteinase inhibitors in a final volume of 10 μL containing the cocktail of proteinase inhibitor cocktail at either 0.1% or 1%, 100 μM E-64, 1 or 10 mM 1,10-phenanthroline (Phen), 10 mM 4-(2-Aminoethyl)benzenesulfonyl fluoride hydrochloride (AEBSF), or 50 μM pepstatin A (Pep). Means with the same letter are not significantly different (*p* < 0.05) at 5 dpi according to Tukey’s test.

**Figure 6 microorganisms-07-00198-f006:**
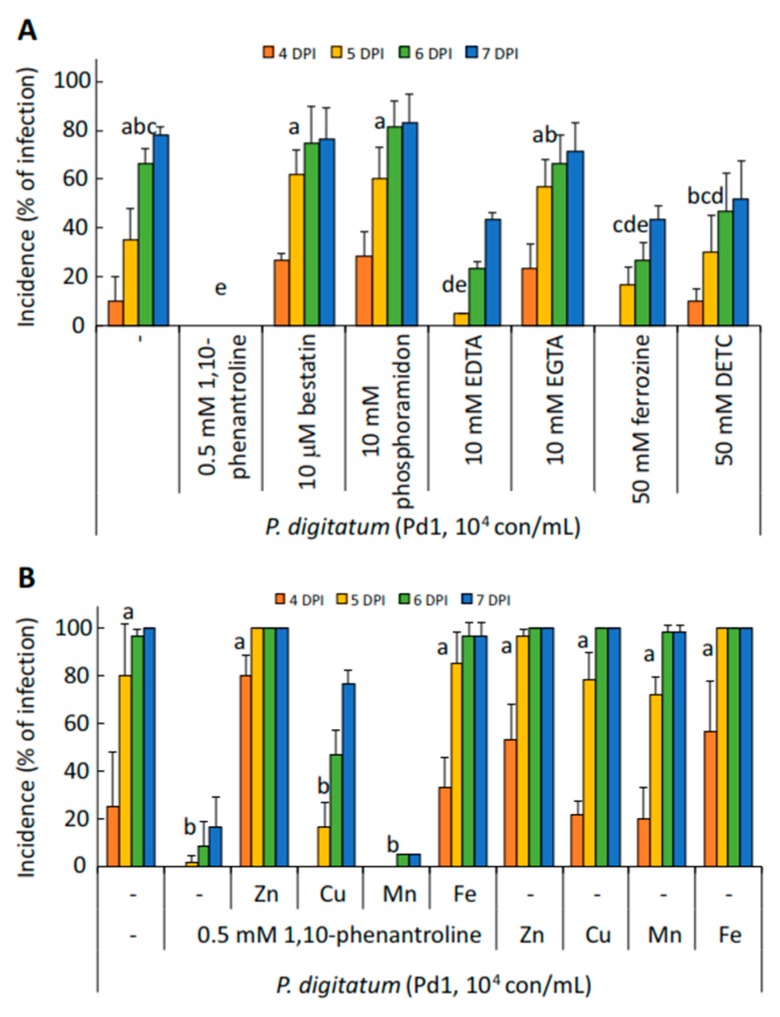
(**A**) Disease incidence (percentage of infection) caused by *P. digitatum* strain Pd1 in artificially inoculated oranges after 4, 5, 6, and 7 days post infection (dpi). Conidia were co-inoculated with different metalloproteinase inhibitors and chelators. (**B**) Disease incidence caused by *P. digitatum* Pd1 co-inoculated with different metal ions (ZnSO_4_, CuSO_4_, MnSO_4_, and FeSO_4_ at 0.5 mM) in the presence or absence of 0.5 mM of 1,10-phenanthroline in mature orange fruit. Means with the same letter are not significantly different (*p* < 0.05) at 5 dpi according to the Tukey’s test.

**Table 1 microorganisms-07-00198-t001:** Primers used in the study. The 5’ deoxyuridine extension parts in the primers used for the assembly of the USER cloning sites are indicated in bold.

Primer Name	Sequence (5’→3’)
**Knockout mutant construction**	
O1	**GGTCTTAAU**TCAACTTGCGTGCTATGATTGAAGGCCT
O2	**GGCATTAAU**TGAGCGAGGACTTTTAGCCAATTGCGA
A3	**GGACTTAAU**TAATTGTCTCGAGCAGATGATGCCTGGG
A4	**GGGTTTAAU**GGTACACTCAGACAGCCGTGGAAGCAAA
**Knockout mutant analysis**	
RF1	AAATTTTGTGCTCACCGCCTGGAC
RF2	TCTCCTTGCATGCACCATTCCTTG
RF5	GTTTGCAGGGCCATAGAC
RF6	ACGCCAGGGTTTTCCCAGTC
HPHPRO4	GCACCAAGCAGCAGATGATA
1F	TATGAGGGGTTGTGGCTTTC
HPHTER2	GCTCCGTAACACCCAATAC
4R	CAAACTCGCAAGAGCCCTAC
5F	TTTGAATCGTGCCACTCACC
6R	ATCGGCATAGCTCCACCAGT
**Determination of T-DNA copy number**	
7F	GCGTTGCATGATTGGTGATG
8R	AGCACAACACAACACCCAAG
PdACTFor2	TGTCACCAACTGGGACGATA
PdACTRev2	GAGCTTCGGTCAAGAGGATG
**Gene expression analysis**	
prtT-F	GATCGTCGCAGAAATCCAAC
prtT-R	TTCCAGCGTTCCAGATCTTC
pep1-F	TGGCTATGTCTTCCCTTGCT
pep1-R	TGACGGAAGCGTAGTTGATG
aor1-F	CTCTGGGCAGCCATTGTATT
aor1-R	TGGTGACTCAAGTGCTCCAT

**Table 2 microorganisms-07-00198-t002:** Estimation of the number of T-DNA copies that have been integrated into the genome of the mutants.

Strain	Genotype	Cq prtT	Cq Actin	ΔCq Target	ΔCq Actin	Copy Number
Pd1	wild type	23.9 ± 0.0	22.1 ± 0.1	0.0	0.0	–
e*prtT*3	ectopic	23.6 ± 0.0	24.9 ± 0.2	0.3	−2.8	6.7
Δ*prtT*44	knockout	23.8 ± 0.1	22.2 ± 0.1	0.1	−0.1	1.1
Δ*prtT*70	knockout	24.5 ± 0.1	22.7 ± 0.2	−0.6	0.0	0.7

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
