# Peer review of "Functional and Pharmacological Analyses of the Role of Penicillium digitatum Proteases on Virulence"

_microorganisms, 2019, doi:10.3390/microorganisms7070198_

Round 1

Reviewer 1 Report

Authors present work on search for factors affecting virulence of P. digitatum. Such data might be critical for understanding and eradication major fungal infections in sweet orange crops and as such they are of general interest to the scientific society.

Numerous studies showed that modulation of virulence in fungal genera is related to overexpression of wide variety of proteases, what makes them a new biological target. In order to identify the target protease authors created a two-fold approach: first they focused on PrtT transcription factor controlling expression of a limited group of proteases and prepared and characterized prtT knockout mutants and second, used a mixtures of inhibitors suppressing activity of different proteases probing level of infection of P. digitatum in sweet oranges. The experimental model for in vitro and in vivo studies involved collection of spores and peel tissue of oranges infected with the parental strain Pd1, the ectopic eprtT3 mutant, and the knockout mutant ΔprtT70.

Authors confirmed that PrtT of P. digitatum is involved in production of limited number of extracellular proteases. However, the absence of this regulator had very little impact on the expression of the genes  encoding the major extracellular proteases secreted by P. digitatum during the infection of orange fruits and its virulence. Second approach was more specific and allowed to identify 1,10‐phenanthroline as an inhibitor with a major role in suppressing disease development. From the known fact that 1,10‐phenanthroline is known inhibitor of metalloproteases, acting by chelation and subsequent removal of the metal ions necessary for the pathogen development its inhibitory activity was reverted by addition of transition metal ions. Among them Zn(II) and Fe(III) were the most efficient.

Manuscript is well written and all statements are sufficiently supported by experimental data as well as supplemented with an extensive referencing to the recent literature. During testing of combination of inhibitors specific for various protease classes authors identified metalloproteases as major factor influencing development of infection and proposed application of 1,10‐phenanthroline - known, relatively cheap inhibitor for its prevention.  

Summing up, authors presented very convincing paper presenting new data on P. digitatum virulence factors in post-harvested sweet orange crops and found possible, relatively simple way to control its development.

I recommend to publish this paper as it is.

Author Response

We appreciate all the good comments of the reviewer. There were no changes required by this reviewer.

Reviewer 2 Report

This manuscript describes an interesting work on proteases in Penicillium digitatum. I do see a solid piece of science worthy of publication in this journal, but also have some concerns that authors should consider to address for the sake of the study.

My major concern is the conclusion and the tone of the whole manuscript. I quote "...this study shows that PrtT, a transcription factor that regulates extracellular proteases, is not the major factor affecting....

The results you included in the manuscript did not indicate the encoded product is indeed a transcription factor. Bioinformatics only allows you to predict functions but is not conclusive, you have to address the prediction with experimentation to confirm it. Experiments such as heterologous complementation of mutants in the same gene, EMSA or DNA fingerprinting should be included to claim such a conclusion. You should change the discussion and conclusion to suggest that is a putative transcription factor.

Minor points

-Again, in the introduction is mentioned a large number of genes encoding for proteases in the genomes of other fungi, but this is a predictive exercise with no experimental proof obtained yet. You should indicate that are genes encoding for putative proteases.

-Was it confirmed that there were no genes overlapping the 1800bp regions used for homologous recombination?

-Please indicate the meaning of the bold letters in table 1.

-Why did the actin gene was used for data normalization in the qPCR reactions? References supporting its use as a gene with no variable expression in this organism should be included.

Author Response

"Please see the attachment

Reviewer 3 Report

Overall, this manuscript is interesting and provides new information on the role of proteases in Penicillium digitatum’s virulence on citrus fruit. And this paper is favor for the development of new alternative treatments to combat postharvest pathogen. However, there are some concerns the authors should addressed before publication.

My major concern is focusing on the first approach. In fact, the first approach can not explain the role of P. digitatum’s proteases in virulence. As the authors discussed in the manuscript, major extracellular proteases secreted by the pathogen during the infection process are not regulated by the prtT gene (Line 491). Why functional characterization of the P. digitatum prtT gene can indicate the role of protease in virulence?? Therefore, the role of prtT can not represents the role of proteases. This will make the readers confused. I think the authors MUST clarify this in the manuscript, especially in the Abstract and Introduction. Maybe, the authors used two approaches to control P. digitatum infection through regulating protease activity. BUT, not two approaches to investigate the role of protease in virulence. From my own opinion, this is different.

My second major concern is the protease activity change inΔprtT knockout mutants. As the authors described “protease activity of the ΔprtT knockout mutants grown on PDA was reduced to almost undetectable levels”. I think the authors should discuss the reason further in the Discussion part.

Finally, it is interesting that the protease activity was significantly reduced in ΔprtT knockout mutants but no influence on the virulence on fruit. Again, I think the authors should give more discussion on the possible reasons.

Minor:

For gene expression, the authors present the data as “relative gene expression”. It is clear that actin was used as housekeeping control gene. But it is not clear what was used as base line for calculating relative expression level. And I did not found the specific primers for gene expression analysis in Table 1.

Line 356, provide the link of SignalIP

Line 436, “the combined application with 1,10phenanthroline substantially reduced disease incidence”, not true. ZnSO4 and FeSO4 together with 1,10phenanthroline did not reduce disease incidence. Please correct it.

Line 537, Why the authors chose these four metal ions? Does protease activity need these metal ions? The authors should explain this in the discussion.

Line 502-505 “The possibility that…” This sentence is too long.

Line 506, Change “In a second approach” to “In the second approach”

Figure 5 and Figure 6 are both disease incidence, why different units for Y axis?

Round 2

Reviewer 3 Report

The revised manuscript has been improved greatly. All of my concerns have been clarified in the manuscript. I agree with the comments on the manuscript made by Reviewer 1 because I am not really an expert in this study field. I suggest that the revised version can be accepted in the present form. 

Author Response

We appreciate the satisfactory comments of the reviewer and the accpetance of the manuscript